# A Systematic Review of the Relationship between Workload and Injury Risk of Professional Male Soccer Players

**DOI:** 10.3390/ijerph192013237

**Published:** 2022-10-14

**Authors:** Zhiyuan Jiang, Yuerong Hao, Naijing Jin, Yue Li

**Affiliations:** 1Sports Coaching College, Beijing Sport University, Beijing 100084, China; 2School of Physical Education, Qingdao University, Qingdao 266071, China; 3Physical Department, Shenzhen Institute of Information Technology, Shenzhen 518172, China

**Keywords:** soccer, football, workload, injury risk, acute:chronic workload ratio, fixture congestion, s-RPE

## Abstract

The number of studies on the relationship between training and competition load and injury has increased exponentially in recent years, and it is also widely studied by researchers in the field of professional soccer. In order to provide practical guidance for workload management and injury prevention in professional athletes, this study provides a review of the literature on the effect of load on injury risk, injury prediction, and interpretation mechanisms. The results of the research show that: (1) It appears that short-term fixture congestion may increase the match injury incidence, while long-term fixture congestion may have no effect on both the overall injury incidence and the match injury incidence. (2) It is impossible to determine conclusively whether any global positioning system (GPS)-derived metrics (total distance, high-speed running distance, and acceleration) are associated with an increased risk of injury. (3) The acute:chronic workload ratio (ACWR) of the session rating of perceived exertion (s-RPE) may be significantly associated with the risk of non-contact injuries, but an ACWR threshold with a minimum risk of injury could not be obtained. (4) Based on the workload and fatigue recovery factors, artificial intelligence technology may possess good predictive power regarding injury risk.

## 1. Introduction

With the development of commercialization and professionalism in soccer, the professional leagues in many countries have long seasons, many games, and congested schedules [1]; further, the game is characterized by a long net time, a fast pace of attack and defense, and intense physical confrontation [2]. All of these factors seem to lead to a high incidence of match injuries among professional players. In order to reduce the risk of injuries in matches and meet the diverse demands of the match, players need to undergo systematic, reasonable physical and tactical training to produce positive physiological adaptations [3]. An insufficient training load does not induce functional adaptation in athletes, and an excessive training load may lead to an increased risk of injury [3], which can affect athletes’ competitive performance and team performance [4,5]. A meta-analysis of the risk of injury in professional male soccer players showed that the overall incidence of injuries was 8.1 injuries/1000 h of exposure, the incidence of match injuries was 36 injuries/1000 h of exposure and the incidence of training injuries was 3.7 injuries/1000 h of exposure [6]. Although sports injuries are caused by the interaction of multiple risk factors, an unreasonable training and competition load is an important external risk factor that can directly or indirectly affect other risk factors [7]. Advantageously, workload can be monitored and manipulated with potential value for injury prevention, compared to non-variable risk factors such as age, sex, and venue. As part of the International Olympic Committee (IOC) consensus statement, experts discussed the relationship between workload (rapid changes in training and competition loads, congested schedules, psychological loads, and travel) and health (injuries and illnesses) [8,9]. Similarly, soccer training and sports medicine researchers have applied a variety of methods to monitor the training and competition loads of professional players for one or more seasons, laying a foundation for the in-depth exploration of the relationship between workload and injury risk [10]. In this scenario, the approach to these relationships has changed, with greater individualization of training for pain management, physical therapy, and rehabilitation, tailored to the athlete [11]. Exploring the right balance between training, playing, and recovery is, therefore, one of the biggest challenges faced by soccer practitioners. This review aims to assess the research results related to the influence of professional male soccer players’ training and game loads on injury risk, determine a safe and efficient level of activity, and provide practical guidance for load management and injury prevention in soccer players.

## 2. Methods

A systematic literature search was conducted according to the Preferred Reporting Items for Systematic Reviews and Meta-Analyses (PRISMA) 2020 statement [12]. Two independent reviewers assessed titles, abstracts, and full-text articles to identify eligibility. If the title and abstract indicated potential inclusion, the full text was reviewed for eligibility. Disagreements between the two reviewers were resolved by including a third reviewer.

### 2.1. Search Strategy

A systematic and disciplined literature review was conducted via several databases including PubMed, Embase, Web of Science, and SPORTEDiscuss. The final Boolean search syntax was TS = (((“football” OR “soccer”) AND (“training load” OR “workload” OR “training volume” OR “load” OR “RPE” OR “sRPE” OR “GPS” OR “global positioning system” OR “match frequency” OR “fixture congestion” OR “congestion” OR “congested” OR “match congestion”) AND (“injury” OR “non-contact” OR “soft tissue” OR “contact” OR “musculoskeletal injury”)) NOT ((“patient” OR “disease” OR “syndrome” OR “cerebral palsy” OR “injury” OR “obese” OR “animals”))). Related research published prior to 31 December 2021 was included.

### 2.2. Selection Criteria

Two reviewers (ZYJ and YRH) selected the articles for full-text appraisal according to the title and abstract. Any disagreement over inclusion was resolved via a discussion between the reviewers. In the case of continued disagreement, a third reviewer (YL) was consulted. The references of all included studies were checked for other relevant articles. Each article had to meet the following criteria: (1) it was an original article written in English, (2) the study population included male soccer players aged ≥18 years, (3) internal and/or external load parameters were described, (4) soccer-related injuries were registered by medical staff or were self-reported, (5) relevant data were reported concerning the effect of workload on injury risk during soccer matches or training. Professional male soccer players were defined as players who participated in the highest competition in their country or internationally. EndNote X7 software (Clarivate Analytics, Philadelphia, PA, USA) was used to perform the selection process.

### 2.3. Data Extraction

The relevant data from each study were extracted: study details (author, year of publication, duration of follow-up, and types of competition), study population (age and sample size), injury definition, and workload (competition workload and training workload). The association between workload and injury risk was directly extracted from the original articles.

Load is usually defined as the cumulative pressure that athletes endure during training or competition over a period of time; it is used interchangeably with workload and training load. Load is mainly quantified in three dimensions: internal load and external load, subjective load and objective load, and absolute load and relative load [13]. The literature generally uses internal load (session rating of perceived exertion, s-RPE) and external load (match exposure; GPS-derived metrics) as monitoring measures, as well as the absolute load and the relative load as calculation methods. The absolute load refers to the sum of loads within a certain time period, mainly considering 2-, 3-, and 4-week loads. The relative load takes into account the rate of load application, the history of loading, or the fitness level of the athlete, and most often expresses variations in loads between two periods. The relative load includes week-to-week changes and the acute:chronic workload ratio. The weekly load variation refers to the difference between the current week’s load and the previous week’s load, and the ACWR is the ratio of the acute load (commonly the prior 7 days) to the chronic load (commonly the prior 3 to 6 weeks). The time window of the ACWR may differ according to the sport being studied and the training schedule [14], and the time window for the chronic load is more likely to be 3 or 4 weeks, which is expressed as ACWR (1:3) and ACWR (1:4).

With regard to the definition of injury, the FIFA Medical Assessment and Research Center’s international consensus statement establishes the definition and methodology to be used in the study of soccer sports injuries, requiring injury incidence to be reported in the number of injuries per 1000 player-hours (P-H) [15]. Sports injury refers to all physical discomfort caused by a soccer match or training, including medical-attention injuries, which result in the athlete receiving medical attention, and time-loss injuries, which result in a player being unable to take full part in future training sessions or match play. The injury severity is defined as the number of days that have elapsed from the date of injury to the date of the player’s return to full participation in team training and availability for match selection.

### 2.4. Methodological Quality Assessment

The quality of each study was assessed using the Newcastle–Ottawa Quality Assessment Scale (NOS) for cohort studies [16]. The NOS is a quality assessment tool that assesses a study in three areas: participant selection (4 items), comparability (1 items), and outcome (3 items). The NOS is one of two tools recognized by the Cochrane Collaboration to assess the methodological quality of non-randomized studies, and it provides a good performance with great ease of use and specific criteria [17]. Each study was evaluated by two authors (ZYJ and NJJ) independently and all discrepancies in scoring were resolved by arbitration between the two reviewers. The level of evidence for each article that met the inclusion criteria was evaluated using the Oxford Centre for Evidence-Based Medicine model [18].

## 3. Results

### 3.1. Study Identification and Selection

Study identification and selection were performed independently by two researchers and included saving the online search, removing duplicates, and screening titles, abstracts, and full texts. A flowchart of the search process and study selection is summarized in Figure 1. An initial 13,031 articles were retrieved from database searches. Duplicate articles were removed, and a further 5689 irrelevant articles were removed based on their title and abstract. A total of 53 full-text articles were screened for eligibility. Finally, 20 met the eligibility criteria and were included in our review (see Table 1) [19,20,21,22,23,24,25,26,27,28,29,30,31,32,33,34,35,36,37,38,39]. All studies were published between 2010 and 2021, but most were published after 2016 (75%).

### 3.2. Methodological Quality

The median overall NOS score was six (range 4–7), the median participant score was three (range 2–4), the median comparability score was one (range 0–2), and the median outcome score was one (range 0–2). The NOS score can be translated to ‘good’, ‘fair’, and ’poor’ levels of study quality as per the US Agency for Healthcare Research and Quality guidelines [12]. The 10 articles that were assessed as ‘good’ were considered to contain level 2b evidence according to the Oxford Centre for Evidence-based Medicine model, while the 10 articles assessed as ‘fair’ or ‘poor’ were considered to contain level 4 evidence [18]. Scores for each article are presented in Table 1.

### 3.3. Study Characteristics

A total of seven studies explored the effects of short-term match congestion [20,21,22,36] and long-term match congestion [20,23,36,40] on the risk of subsequent injury in professional players. However, the incidence of match injuries was significantly higher when there were ≤3-day intervals [21] and ≤4-day intervals [19,20,22] between the two matches compared to ≥6-day intervals. Two studies concluded that long-term match congestion did not have an impact on the overall incidence of injuries or muscle injuries [24] and had no relationship with the incidence of match injuries [23]. The results of the analysis are presented in Table 2.

The total distance, high-speed running distance, sprint distance, number of accelerations and deceleration in GPS variables, and the weekly cumulative load and ACWR are all used to discuss the relationship between external load and injury risk. The results of the analyses are presented in Table 3. As clearly summarized in Table 4, a total of eight studies discussed the relationship between the ACWR (1:3 and 1:4) and injury risk in soccer players. Five of these studies supported a medium-to-large association between the ACWR and the non-contact injury risk in elite professional soccer players [10,30,32,33,41]. Two studies of elite young players demonstrated that the ACWR was not associated with the risk of non-contact injury [34,35]. The correlation between the weekly cumulative load of s-RPE and the risk of non-contact injury cannot be determined [10,35,42]. In four studies using s-RPE to predict non-contact injuries in soccer players, the AUC values of the weekly cumulative load, the weekly load change, and the ACWR (1:3 and 1:4) were all 0.4–0.6 [10,30,32], the predictive power was poor, and none of them could be used alone as a prediction tool for non-contact injuries in soccer players.

## 4. Discussion

### 4.1. Relationship between Game Load and Injury Risk in Soccer Players

#### 4.1.1. Effect of Competition Calendar Congestion on Injury Risk in Soccer Players

Exposure refers to the number of times or duration of a player’s participation in a competition monitored during competitions with other teams. It is the most basic method for monitoring external load. European professional teams currently play 50–80 matches in a ~40 week season, thus regularly playing two matches per week, with some completing as many as three matches in a weekly microcycle [23]. The increase in the number of athletes participating in competitions for a short or long period is called fixture congestion or match congestion. The precise definition of a dense schedule is when the recovery time between at least two games is less than 96 h, and its impact on players’ match performance and risk of injury has been widely studied. Short-term match congestion refers to the number of days between players participating in two consecutive matches, and long-term match congestion refers to the total length or frequency of the athlete’s exposure to matches in the past 20 days.

Short-term match congestion causes soccer players to have an increased risk of match injury, which is related to the specific interval between the two matches. Bengtsson et al. argue that there were no differences in overall, muscle or ligament incidence rates between matches played with three or less days recovery compared with matches with four or more days recovery, in any competition [20]. Similarly, the former did not lead to a higher incidence of match injury and the severity of the injury rate when compared with ≤3-day and ≥4-day intervals [22]. A study by Dupont et al. showed that the overall injury incidence of two matches per week with a 3–4-day interval is six times that of a match per week [19]. In addition, short, congested match cycles also had an influence on the timing distribution of match injuries, and the injury risk was especially high in the final 15 min of the second match of a two-match congestion cycle and in the first-half of the final game in a three-match congestion cycle [21]. The higher risk of injury caused by 3–4-day intervals between mathes, probably because the fatigue level of players is higher than a single match in a week. The English Premier League players played 4 days apart from one another, and creatine kinase levels and fatigue levels before the second match were significantly higher than those found when they were playing a match per week [43]. Athletes’ subjective health, overall quality of recovery, and adductor strength required at least 96 h to recover in the event of more than one match per week [44]. These results are consistent with the IOC’s recommendation that matches should be interspersed by at least 96 h to protect players from injury [8]; however, this recommendation has still not been taken in consideration by soccer governing bodies.

Long-term match congestion may not affect the overall injury incidence and match injury incidence. Dellal et al. argued that a long-term dense schedule did not have an impact on the overall injury incidence, but the incidence of race injuries increased significantly, while the incidence of training injuries decreased [1]. In addition to the impact on the incidence of injuries, the severity of injuries caused by long-term match congestion is lower than that of non-congested periods [23]. These conclusions are attributable to the active implementation of the rotation strategy and post-match recovery strategy by the teams [23], as well as the self-adjustment of training and competition loads by athletes.

In essence, short-term match congestion causes a rapid increase in match exposure over a short period, while long-term match congestion causes cumulative fatigue in players after prolonged match exposure. Dense schedules not only just about increasing 90-min activity when players were inadequate recovery, but also engender a lack of motivation and both physical and mental exhaustion caused by factors such as mental preparation, jet lag, and climatic adaptation [40]. These aspects further contribute to their declining decision-making, attention, and coordination skills in a match and put them at a high risk of injury. Another mechanism of explanation is that previous competitions may have caused a decrease in the athlete’s sprinting ability, maximum strength, and jumping ability, which have a passive impact on the biomechanical structure of the lower limbs [45]. A meta-analysis of the match-running performance of professional male players confirmed that their total distance was unaffected, but their low-intensity and moderate-intensity running distances decreased during congested schedules, while players still maintained high-intensity running distances by employing pacing strategies. This means that players who maintain the same high intensity loads in non-congested periods under the influence of fatigue are more likely to experience sports injuries.

#### 4.1.2. Effect of Cumulative Match Exposure on Athletes’ Risk of Competition Injury

Although the number of matches played in different national leagues varies, elite teams and players tend to participate in higher numbers and types of matches, with higher cumulative match exposure. Players who participated in the World Cup in Korea and Japan completed more games than those who did not (46 vs. 32), but there was no significant difference in the overall injury rate and match injury rate between the two. Moreover, no difference was observed in the incidence of match injuries between high-match-exposure players and low-match-exposure players (18.6 vs. 14) from elite South American clubs, but injuries were more severe in high-match-exposure players.

Similarly, the shortening of the winter break in professional leagues has led to an increase in the incidence of injury. After the German Bundesliga’s winter break was shortened from 6.5 weeks to 3.5 weeks, there was no significant difference in the overall incidence of injuries in the second half of the league, but the incidence of serious injuries in the game (after a 7-day absence) was higher. The above results are also supported by Ekstrand et al. [46], who believes that teams without a winter break (English clubs) had a higher incidence of severe injuries(absence of more than 28 days) following the time of the year that other teams (other European clubs) had their scheduled break. Therefore, increased cumulative match exposure does not change the incidence of match injuries but leads to a higher incidence of severe injuries. In view of the influence of cumulative match load on injury risk, detailed monitoring and planning of training and competition, as well as developing systematic and effective injury prevention programs and intervention strategies, may help to reduce the serious injuries caused by increased match load and the shortening of winter breaks.

### 4.2. Relationship between Training Load and Injury Risk in Soccer Players

#### 4.2.1. Effect of GPS-Derived Metrics on Injury Risk

Over the past 10 years, GPS and accelerometer technologies have become the main external load-monitoring tools for team sports, and can objectively and accurately quantify the distance and speed of training and competition, providing a basis for researchers to explore the relationship between GPS variables and injury risk. Training and match exposure provide only rough statistics regarding external load, while load variables such as running distance, speed, change of direction, and acceleration are more explanatory in relation to the risk of injury to athletes [47]. As can be seen in Table 3, the total distance, high-speed running distance, sprint distance, number of accelerations and decelerations in GPS variables, and the weekly cumulative load and ACWR are all used to discuss the relationship between external load and injury risk.

The higher the weekly cumulative load at the total distance, the higher the risk of overuse injury [48]. Compared with 0.88–1.11, the overuse injury ratio was higher when the ACWR (1:4) of the total distance was <0.88 and >1.11 [38]. Similarly, a comparison of the total distance between injured and uninjured players revealed that the risk of contact injury was highest at an ACWR (1:4) = 1.76, and a gradual increase in chronic load established a tolerance to high acute loads [25]. In addition, a significant increase in the intensity of training and competition led to an increased risk of injury. Players’ relative distances were 9.6% and 7.4% higher than the season average in the first week and the first four weeks before injury, respectively.

High-speed running distances (>20 km/h) at an ACWR > 1.18 resulted in a higher risk of overuse injury risk [38]. The risk of non-contact injury increased significantly when high acute (1-week) and low chronic (4-week) high-speed running (>20 km/h) distances were combined, and the risk of non-contact injury was not significantly increased when high acute and high chronic high-speed running distances were combined [25]. These two studies suggest that a high ACWR combined with a low chronic load leads to a higher risk of injury. On the other hand, a weekly load that is too high or too low leads to an increased risk of injury for athletes. The moderate (701–750 m) high-speed running distance (>14.4 km/h) and the medium (201–350 m) sprint-running distance (>19.8 km/h) in a week were lower than those at low levels [27]. Similarly, high-sprint distance training weeks resulted in an increased risk of non-contact injury, and the time interval between new injuries and the previous injury was shorter than that of low-load weeks [49]. In addition, only Enright et al. demonstrated no correlation between total distance, high-speed running distance, and sprint distance, and non-contact injury type (muscle, ligament, and tendon) or severity of injury [28].

For the number of accelerations in training, the greatest risk of contact injury occurs when there is an ACWR (1:4) > 2 and a chronic load < 1731 [29] and an ACWR (1:4) of 1.77 leads to the greatest risk of contact injury [25]. At low 4-week loads, an ACWR > 2 leads to a 5- to 6-fold higher risk of non-contact injury than an ACWR < 2 [29]. These two studies reflect that a rapid increase in the number of accelerations under acute loads leads to an increased risk of injury, especially in the case of low chronic loads. In addition, the 3-week cumulative load (>9254 times) was most correlated with the overall risk of injury and the risk of non-contact injury [25]. Only one study had explored the relationship between deceleration and overuse injury risk, the study showed that an increased injury risk for higher 2- to 4-week loads, and a lower injury risk was found for a medium ACWR = 0.86-1.12 than ACWR < 0.86 and ACWR > 1.12 [38].

In summary, the total distance, high-speed running distance, and number of accelerations all increased the risk of injury at high ACWRs and high weekly cumulative loads, but it is not possible to draw accurate conclusions about their relationship with injury risk. On the one hand, there are few studies on the application of GPS to explore injury risk in elite athletes, so it is difficult to compare the results. On the other hand, there are differences in injury definitions and types, statistical analysis methods, speed threshold settings, and GPS devices (manufacturer, specification, and sampling frequency) among different studies. During training, there should be a focus on monitoring the total distance, high-speed running distance, and acceleration frequency of athletes, and a large amount of the load should be maintained in the form of medium-to-large long-term loads when arranging training and match loads, while avoiding a rapid increase in acute loads.

#### 4.2.2. Effect of s-RPE on Injury Risk

The s-RPE (session RPE) is the measure of quantitative internal load proposed by Foster on the basis of RPE. The training load is the RPE value 15~30 min after a training session or a game multiplied by the exercise duration (minutes) in AU (arbitrary units) [50]. In general, the loads of a low-intensity and high-intensity soccer training session are 300–500 AU and 700–1000 AU, respectively. Compared with other traditional monitoring methods, s-RPE can effectively explain the unstable pressure of athletes in mixed training sessions (including technology, tactics, and physical fitness) [51], with the advantages of low cost, high efficiency, reliability, and limited data loss [52]. As one of the most commonly used subjective internal load measures, s-RPE has been widely applied in correlation studies of non-contact injuries in soccer.

##### Effect of ACWR on Non-Contact Injury Risk

The combination of s-RPE and the ACWR was the most numerous in studies on the relationship between workload and injury in soccer players. The ACWR is not an indicator of chronic undertraining or overtraining, but rather an indicator of assessing acute load changes. Chronic workload in the ACWR represents the state of adaptation, acute workload represents the state of fatigue, and the comparison of acute load to chronic load reflects the state of readiness of the athlete. In 2016, the IOC recommended using ACWR to monitor injury risk and provided thresholds to minimize risk when designing training programs [8]. In recent years, however, some researchers have pointed out several limitations to the ACWR and have posited that how it has been analyzed impacts the validity of current recommendations, indicating that we should discourage its use. Wang covered problems with discretization, sparse data, bias in injured athletes, unmeasured and time-varying confounding, and application to subsequent injuries when the ACWR was implemented [53]. The relation between the ACWR and injury risk are not supported by etiology theory [54]. At the same time, there is a lack of background rationale to support its causal role, it is an ambiguous metric, and it is not consistently and unidirectionally related to injury risk [55].

In addition to the significant correlation between the ACWR and non-contact injury risk, researchers have attempted to explore the ACWR threshold for minimizing the risk of injury and have provided specific recommendations for coaches to reasonably arrange athletes’ workloads. In terms of injury risk, an ACWR within the range of 0.8–1.3 could be considered the training ‘sweet spot’, while an ACWR ≥ 1.5 represents the ‘danger zone’. To minimize injury risk, practitioners should aim to maintain the ACWR within the range of approximately 0.8–1.3. It is possible that different sports will have different training load–injury relationships [56], even if, in soccer, it is hard to infer a consistent ACWR threshold with a low injury risk. Malone et al. argue that the injury risk is lower when the ACWR = 1.00–1.25 than when the ACWR < 0.85 [41], Delecroix et al. show that the injury risk is lower when the ACWR is > 0.85 than when the ACWR is <0.85 [30], and Jasper et al. show a U-shaped relationship between them, with the injury risk being lowest when the ACWR = 0.85–1.12 [38]. Three other studies reported an increased risk of injury with an increase in ACWR. Fanchini et al. argue that the higher the ACWR, the higher the risk of non-contact injury, and that there is no protective zone [32]. Similarly, Tiernan et al. report that the non-contact injury risk increases at an ACWR > 1.2 [33]. McCall et al. report that the risk of injury when the ACWR = 0.6–0.97 is lower than that when it is in the range of 0.97–1.38 and when it is >1.38. The ideas and results of the above studies vary greatly, and the relationship between the ACWR and injury risk is both direct and inverse; thus, it is therefore impossible to infer a consistent ACWR threshold with a low risk of injury, which means that a training load with a low risk of injury cannot be determined [10].

##### Effect of Weekly Cumulative Load on Non-Contact Injury Risk

The correlation between the weekly cumulative load of s-RPE and the risk of non-contact injury cannot be determined. McCall et al. concluded that weekly load was not significantly associated with non-contact injuries [10], and another study of elite young professional players showed that weekly load was not associated with non-contact injuries [42]. Delecroix’s findings for players of different ages varied, with weekly cumulative load in U19 athletes not independent of the overall incidence of injury, and weekly cumulative load in U21 athletes positively correlated with injury risk [35]. Unlike the results of the three studies mentioned above that investigate full seasons, two studies on the pre-season period concluded that weekly loads were associated with non-contact injuries. The risk of contact and non-contact injuries at 1500–2120 AU was significantly higher than at <1500 AU under a pre-season weekly load [41]. In another pre-season study, both 3- and 4-week cumulative loads were associated with the incidence of non-contact injuries [30]. The main reasons may be that athletes experience decreased physical fitness after the offseason, and the excessive cumulative load in the early season causes soft tissue injuries [56]. In addition, more pre-season training by several European teams was not associated with the incidence of injuries during the season, but 10 more training sessions reduced the incidence of serious injuries in matches by 0.18 injuries/1000P-H and increased match attendance by 1% [57]. This suggests that excessive cumulative loads early in the season can lead to an increased risk of injury, while gradually increasing the training load and avoiding a rapid increase in load can have a protective effect during the season.

### 4.3. Soccer Player Injury Prediction Based on Workload

When discussing the relationship between workload and injury, the terms ‘correlation’ and ‘prediction’ are often used interchangeably, resulting in correlations often being misunderstood as predictors of injury. Injury-related load factors are used to identify athletes at increased risk of injury and to justify injury prevention strategies, while the load factors associated with injury prediction are those that determine the athletes who will be injured [10]. Injury prediction involves predicting whether an athlete under a recent training load will be injured in the next training session or official match, thus providing effective support for team decision-making. Predicting injury risk through workload has always been a concern in the field of soccer injuries. Another study mainly tested the predictive power of subjects according to the area under the curve (AUC) of the receiver operating characteristic curve (ROC). AUC = 1 represents perfect predictive power, AUC = 0.5 represents no predictive power, and AUC = 0.7 is the reference point for good injury prediction power [58]. The main reason for the poor predictive power is that sports injuries are generated by the interaction of multiple risk factors, and accurate prediction by only one or more measures under the load dimension is too oversimplified to fully understand the physical state of the athlete.

Predictive analysis is based on the ability to predict future outcomes based on historical data, and artificial intelligence (AI) has an advantage in managing individualized diverse data. Therefore, in recent years, some researchers have begun to try to use AI to predict injuries in soccer players. Mandorino et al. used s-RPE and total recovery quality to quantify the internal training/game load and recovery status of soccer players, respectively, and calculated the cumulative load (2, 3, and 4 weeks) and ACWR (1:4). The classification tree model displayed good discrimination (AUC = 0.76), low recovery status, a rapid increase in training load, and cumulative load and maturity were identified by data mining algorithms as the most important injury risk factors [59]. Vallance et al. used GPS data and subjective questionnaire data (sleep quality, fatigue, etc.) to apply a variety of classification machine learning algorithms and found that the internal loads were more accurate than 1-week injury prediction external loads, while for 1-month injury prediction, the best performances by the classifiers were reached by combining internal and external loads [60]. Mandorino et al. believe that the support vector machine (SVM) is the best prediction tool for the risk of muscle injury in young soccer players (AUC = 0.84) and use the decision tree algorithm to understand the interactions identified by the SVM model, assessing how the risk of injury could change according to players’ maturity status, neuromuscular fatigue, anthropometric factors, higher workloads, and low recovery status [61]. In addition to the application of AI, the above prediction model adds fatigue recovery factors to the workload and injury risk in order to establish a causal relationship between load, fatigue recovery, and injury risk, which helps to determine the reasonability of specific injury risk factors and load measures. If fatigue recovery factors are ignored, adjusting the workload may avoid the injury risk, but it will also constrain the achievement of training goals and the improvement of athletes’ match performance. Considering the multi-factorial nature of sports injury occurrences, more high-quality parameters (external load, heart rate, and sleep quality) should be collected, the sample size should be increased (multiple teams and multiple-season training data), and a reasonable prediction model should be selected, which may help to build a prediction system for injury risk.

### 4.4. Mechanism Analysis Based on Workload Injury Etiology Model

The relationship between the workload of soccer players and injury risk cannot be clearly defined. In addition to the common problems of concept definition differences and research method flaws in the evaluated studies, another major problem is the lack of a conceptual framework or reference models to accurately interpret the current research results. With the deepening of epidemiological research on sports injuries, researchers have been trying to explain the impact of workload on injury risk and integrate them into various sports injury theory models. Windt and Gabbett [62] proposed a workload–injury etiology model based on previous research, which better explains the dynamic and cyclical effects of load on sports injuries. The load in the model is seen as a medium for the causal chain of injury, exposing the athlete to external risk factors such as equipment and the environment, as well as potential inciting events, such as cumulative overload and falls, and does not cause injuries as a direct risk factor. However, the long-term and repeated application of loads to the athlete has both positive training effects (e.g., aerobic capacity and strength gains) and negative training effects (e.g., decreased neuromuscular control) on the modifiable internal risk factors [63]. The model mainly explains the relationship between load and injury in three ways: First, excessive training and competition load expose athletes to external risk factors and potential inciting events, and the risk of injury increases. Second, negative changes to modifiable internal risk factors induced by workloads increase injury risk. Third, reasonable loads maximize positive effects and minimize negative effects, as well as making athletes more resistant to injury.

As a team-based sport, the training and competition loads of soccer are uniformly arranged, but the application of the same load will have different effects on the modifiable risk factors of players with different fitness levels, and will then produce different degrees of injury risk. This is also confirmed by the results of a series of studies related to soccer injuries. Players with increased intermittent aerobic capacity were better able to tolerate increased absolute changes in training load than players with lower fitness levels [41]. Improvements in VO_2_max during the pre-season training period were significantly lower among injured players (0.9 ± 5.5%) compared to non-injured players (10.4 ± 6.5%, *p* < 0.05) [36], and well-developed lower-body strength, repeated-sprint ability, and speed were associated with better tolerance to higher workloads and reduced risk of injury in team-sport athletes [37]. When a player does not perform well in a physical fitness test during the pre-season, their training load should be lowered slightly and an individual program should be provided to improve their fitness level. When a player has good fitness test scores and no history of injury, their training load should be moderate to high in order to improve their technical and tactical capability without reducing their physical fitness.

## 5. Conclusions

All of the studies discussed in this review take male professional elite soccer players as their research objects, and the following conclusions can be drawn: (1) It appears that short-term schedule congestion leads to an increase in the incidence of match injuries, while long-term schedule congestion has no effect on the incidence of overall injuries and match injuries. (2) It is impossible to determine conclusively whether any global positioning system (GPS)-derived workload metrics (total distance, high-speed running distance, and accelerations) are associated with an increased risk of injury. (3) It seems that the ACWR of s-RPE was significantly correlated with the risk of non-contact injury, while the ACWR threshold with the lowest risk of injury could not be determined. (4) Based on the training load and fatigue recovery factors, the use of AI may have a good predictive effect on injury risk.

Future studies should focus more on the following aspects: (1) There are still some differences in the definition of injury types and severity in current studies, and overall injuries should include all types of injuries in the study. Future studies should select more homogeneous injured players, according to more specific injury definitions, and combine the injury mechanism to clearly understand the quantification of the relationship between workload and a certain type of injury. (2) The current study duration ranges from a certain stage of the season (training camp, pre-season, or mid-season) to one or more seasons, and the number of subjects and the number of injuries are both small, which affects the effectiveness of the statistics. Future studies should conduct large-sample, multi-team, and multi-season research with complete records and reasonable statistics to improve the scope of the results and recommendations. (3) Given many current challenges, some scholars have called for a consensus metting in the field to provide evidence-based recommendations on the monitoring of training and match load in professional soccer [64]. Simple measures such as ACWR and cumulative weekly load are becoming increasingly popular as soccer sports load monitoring methods, and discussing their relationship with specific injury types should take into account cumulative tissue damage, mechanical load, psychological–physiological fatigue, and recovery, which may contribute to a clear understanding of load–injury relationships. (4) In the future, the relationship between load and injury risk in high-level youth soccer players should be explored. The availability to train and gain match exposure are likely essential for high-level young soccer players to develop their physical fitness, and to improve the technical and tactical skills necessary for adult soccer players [65]. Sustaining an injury reduces a player’s availability and may also impair the progress of their development and future career opportunities [65].

## Figures and Tables

**Figure 1 ijerph-19-13237-f001:**
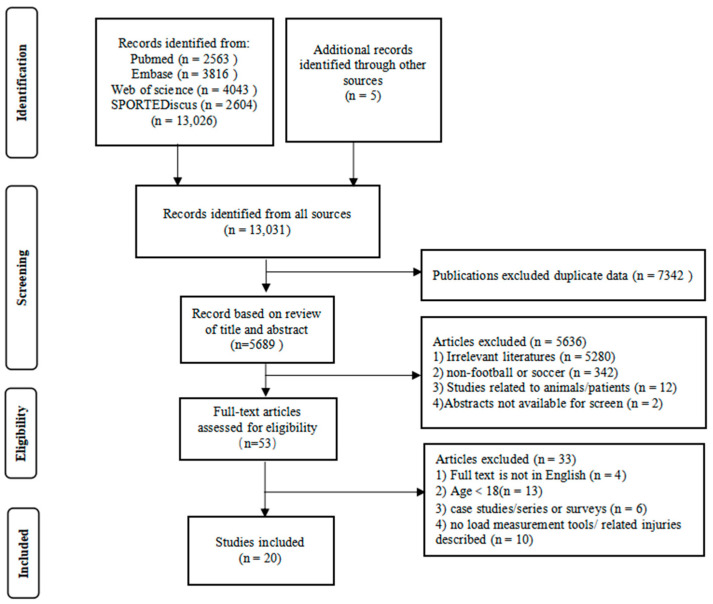
Flowchart of the search process and study selection.

**Table 1 ijerph-19-13237-t001:** Quality of included articles as assessed on the NOS.

Study (Author, Year)	NOS Score				Level of Evidence
	Selection	Comparability	Outcome	Total Score	
Carling et al., 2010 [22]	2	1	1	4	4
Dupont et al., 2010 [19]	3	1	1	5	4
Carling et al., 2012 [23]	3	0	1	4	4
Bengtson et al., 2013 [20]	3	2	2	7	2b
Dellal et al., 2013 [36]	3	1	1	5	4
Carling et al., 2016 [21]	4	1	1	6	2b
Bengtsson et al., 2018 [24]	4	2	1	7	2b
Bowen et al., 2016 [25]	3	2	2	7	2b
Bowen et al., 2019 [29]	3	2	2	7	2b
Ehrmann et al., 2016 [26]	2	1	1	4	4
Malone et al., 2018 [37]	3	2	1	6	2b
Jaspers et al., 2017 [38]	4	1	0	5	4
Enright et al., 2020 [28]	3	1	2	6	2b
Malone et al., 2017 [27]	3	1	1	5	4
McCall et al., 2018 [39]	4	1	2	7	2b
Delecroix et al., 2018 [30]	2	1	2	5	4
Fanchini et al., 2018 [32]	2	1	1	4	4
Delecroix et al., 2019 [35]	2	1	1	4	4
Raya-Gonzal et al., 2019 [34]	3	1	2	6	2b
Tiernan et al., 2020 [33]	3	1	2	6	2b
Median (range)	3 (2–4)	1 (0–2)	1 (0–2)	6 (4–7)	-

2b: Low-quality randomized controlled trial or cohort study with good reference standards. 4: Case series or poor-quality prognostic cohort study.

**Table 2 ijerph-19-13237-t002:** Research literature on the relationship between fixture congestion and injury risk.

Author, Time	Participants	Follow-Up	Injury Definition	Results
Carling et al., 2010 [22]	N = 31, professional UEFA team	Prospective study, four seasons	Time-loss injuries	A very short interval (≤3 days) between fixtures did not result in a greater injury rate or number of days lost to injury compared to a longer interval (≥4 days).
Dupont et al., 2010 [19]	N = 32, top-level team participating in the UEFA Champions League	Retrospective study, two seasons	Time-loss injuries	The injury incidence was significantly higher when players played two matches per week (3–4 days interval) versus one match per week (25.6 injuries/1000 P-H vs. 4.1 injuries/1000 P-H; *p* ≤ 0.01).
Carling., 2012 [23]	N = 26, professional soccer team in French League	Perspective study, eight successive official matches over 26 days	Time-loss injuries	The match injury incidence during congested fixture periods was similar to rates outside these periods (50.3 injuries/1000 P-H vs. 49.8 injuries/1000 P-H, *p* = 0.940), but the mean lay-off duration of injuries was substantially shorter during the former (2.0 ± 1.5 vs. 7.9 ± 14.6 days, *p* = 0.043).
Bengtsson et al., 2013 [20]	Number of participants not indicated, 27 European professional teams from 10 countries	Perspective study, 11 seasons	Time-loss injuries	There were no significant differences in the incidence of total injuries, muscle injuries and ligament injuries compared with intervals of ≤3 days and >3 days between matches; the total injury incidence and muscle injury incidence were increased in matches with intervals of ≤4 days compared to those with intervals of ≥6 days, especially hamstring and quadriceps injuries.
Dellal et al., 2013 [36]	N = 22, professional team in French League	Perspective study, three different congested periods of matches during one season	Time-loss injuries	The overall injury incidence (matches and training) in the congested period (6 matches in 18 days) did not differ significantly from the non-congested period. The match injury incidence was significantly higher during the congested period compared with the non-congested period (*p* < 0.001). The injury incidence during training was significantly lower during the congested period compared with the non-congested period (*p* < 0.001). The mean lay-off duration for injuries was shorter during the congested period compared with the non-congested period (9.5 ± 8.8 days vs. 17.5 ± 29.6 days).
Carling., 2016 [21]	N = 25, professional team in French League	Prospective study, six seasons	Time-loss injuries	In two-match cycles (≤3-day intervals), the injury incidence showed that there was a higher risk of injury in the final 15 min of play in the second match in comparison to matches outside these cycles. A greater risk of injury overall and in the first half of play, and a greater risk of ankle sprains and non-contact injuries were observed in the final match of three-match congestion cycles (≤4-day intervals) in comparison to matches outside these cycles.
Bengtsson et al., 2018 [24]	N = 2672, 57 professional European teams from 16 countries	Prospective study, 14 seasons	Time-loss injuries	No difference in total match injury rates were found between the reference category (≤3 days) and the other categories (4, 5, 6 and 7–10 days) of short-term congestion. Muscle injury rates were significantly lower in matches preceded by 6 or 7–10 days compared with ≤3 days since the last match exposure. No differences in total and muscle injury rates between the three long-term match congestion groups were found (≤4.5 h, 4.5–7.5 h, >7.5 h of match exposure hours in the 30 days preceding a match).

**Table 3 ijerph-19-13237-t003:** Research literature on the relationship between GPS indicators and injury risk in soccer players.

Author, Time	Participants	Follow-Up	Injury Definition	Results
Bowen et al., 2016 [25]	N = 32, mean age: 17.3 ± 0.9 years, English Premier League U18-23 teams	Prospective study, 2 seasons	Time-loss injuries	A very high numbers of acceleration (>9254) over 3 weeks was associated with the highest significant overall and non-contact injury. Non-contact injury risk was significantly increased when a high acute HSR distance (>20 Km/h), but not with high chronic HSR distance. Contact injury risk was greatest when ACWR TD and ACC were very high (1.76 and 1.77, respectively).
Bowen et al., 2019 [29]	N = 33, mean age: 25.4 ± 3.1 years, English Premier League team	Prospective study, 3 seasons	Time-loss injuries	The greatest non-contact injury risk was when the chronic exposure to DEC was low (<1731) and the ACWR (1:4) was >2. Non-contact injury risk was also 5–6 times higher for accelerations and low-intensity distance when the chronic workloads were categorized as low and the ACWR (1:4) was >2, compared with ACWRs below this.
Ehrmann et al., 2016 [26]	N = 19, mean age: 25.7 ± 5.1 years, Australian League team	Retrospective study, 1 season	Time-loss injuries	Players performed significantly higher meters per minute in the weeks preceding an non-contact injury compared with their seasonal averages (+9.6 and +7.4% for 1- and 4-week blocks, respectively), indicating an increase in training and gameplay intensity leading up to injuries. Furthermore, injury blocks showed significantly lower average new body load compared with seasonal averages (−15.4 and −9.0% for 1- and 4-week blocks, respectively).
Malone et al., 2018 [37]	N = 37, mean age: 25.3 ± 3.1 years, Portugal league team	Prospective study, 1 season	Time-loss injuries	Players who completed moderate HSR (701–750 m) and SR distances (201–350 m) were at reduced injury risk compared to low HSR (≤674 m) and SR (≤165 m) reference groups. Injury risk was higher for players who experienced large weekly changes in HSR (351–455 m) and SR distances (between 75–105 m). Players who exerted higher chronic training loads (≥2584 AU) were at significantly reduced risk of injury when they covered 1-weekly HSR distances of 701–750 m compared to the reference group of <674 m.
Jaspers et al., 2017 [38]	N = 35, mean age: 23.2 ± 3.7 years, Dutch League	Prospective study, 2 seasons	Time-loss injuries	For cumulative loads, results in indicated an increased injury risk for higher 2- to 4-weekly loads as indicated by TD, DEC. For ACWR, a high ratio for HSR distances (>1.18) resulted in a higher injury risk. In contrast, a lower injury risk was found when comparing medium ratios for ACC (0.87–1.12), DEC (0.86–1.12)
Enright et al., 2020 [28]	N = 192, age not indicated, UEFA League team	Retrospective study, 2 seasons	Time-loss injuries	The weekly cumulative load (1, 2, 3, 4 weeks) and ACWR (1:3 and 1:4) of the total distance, high-speed distance (>5.5 m/s), sprint distance (>7.0 m/s) did not differ significantly from the risk of non-contact injury, and there was no significant correlation between GPS indicators and the severity of injury; There were no differences in accumulated weekly loads and ACWR calculated by TD, HSR distance, SR distance between muscle, ligament, and tendon injuries. Correlation between each workload variable and injury severity highlighted no significant association.

HSR: high speed running; SR: sprint running; AU: arbitrary unit. TD: total distance; DEC: deceleration; ACC: acceleration.

**Table 4 ijerph-19-13237-t004:** The relationship between ACWR for s-RPE and injury risk in the research literature.

Author, Time	Participants	Follow-Up	Injury Definition	Result
Malone et al., 2017 [27]	N = 37, mean age: 25.3 ± 3.1 years, Portuguese league team	Prospective study, one season	Time-loss injuries	Players who had an in-season ACWR of >1.00 to <1.25 were at significantly lower risk of injury compared to the reference group.
McCall et al., 2018 [39]	N = 171, mean age: 25.1 ± 4.9 years, five elite European teams	Prospective study, one season	Time-loss injuries	A greater risk of non-contact injury was found for players with an ACWR (1:4) of 0.97 to 1.38 and >1.38 compared with players whose ACWR was 0.60 to 0.97. An ACWR (1:3) of >1.42 compared with 0.59 to 0.97 displayed a 1.94 times higher risk of non-contact injury.
Delecroix et al., 2018 [30]	N = 130, five European domestic and confederation-level teams	Prospective study, one season	Time-loss injuries	Non-contact injury incidence was higher when the ACWR (1:4) was <0.85 versus >0.85 and with an ACWR (1:3) >1.30 versus <1.30.
Fanchini et al., 2018 [32]	N = 34, mean age: 26 ± 5 years, professional Italian team	Prospective study, three seasons	Time-loss injuries	Injury risk increased when a player had an ACWR (1:2) of 1.00–1.20, >1.20 compared to <0.81. Injury risk increased when comparing an ACWR (1:3) of 1.01–1.23, >1.23 vs. <0.80. Injury risk increased when comparing an ACWR (1:4) of 0.78–1.02, 1.02–1.26, >1.26 vs. <0.78.
Delecroix et al., 2019 [35]	N = 122, mean age: U19 (16.8 ± 0.9 years), U21 (20.1 ± 0.3 years), elite academy team and professional soccer team competing in first French League	Prospective study, five seasons	Time-loss injuries	There was no association between ACWR and contact, non-contact, and overall injury incidence in U19 and U21 players.
Raya-Gonzales et al., 2019 [34]	N = 22, mean age: 18.6 ± 0.6 years, Spanish First Division U19 Championship team	Prospective study, one season	Time-loss injuries	ACWR was not associated with the non-contact injury rate in the subsequent week.
Tiernan et al., 2020 [33]	N = 15, mean age: 23.4 ± 4.8 years, Irish Premier League team	Prospective study, one season	Time-loss injuries	An increase in ACWR (>1.2) was associated with an increased risk of both a contact and non-contact injury 5 days later.
Jaspers et al., 2019 [38]	N = 35, mean age: 23.2 ± 3.7 years, Dutch league teams	Prospective study, two seasons	Time-loss injuries	A lower injury risk was found when comparing medium ACWRs for s-RPE (0.85–1.12) to low ratios.

## Data Availability

All data generated or analyzed during this study are included in this published article.

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
