# Peer review of "A Systematic Review of the Relationship between Workload and Injury Risk of Professional Male Soccer Players"

_ijerph, 2022, doi:10.3390/ijerph192013237_

Round 1
Reviewer 1 Report
The review has a hint of systematic, if so, insert a methods section in the abstract
Moreover, as repeated at the end of the manuscript. This review suggests .. it appears, it seems .. these definitive conclusions have been softened because there are few articles, among other things recently with the pandemic and with the resumption of the post-lockdown championships there has not been this wave of injuries as it was expected
23 put football because it can enrich the dissemination of the manuscript
30 I recommend more than 2 bibliographic references for such full-bodied statements. Is this number of injuries really greater?
32 here too, have they really modified training management to address these concerns? If so, please provide bibliographic references.
37 This sentence can serve as a basis for the rationale for the study objective .. move it to the end of the introduction
49 Maybe I can suggest a rationale like this: “In this scenario, the approach to these relationships has changed, with greater individualization of training, for pain management, physical therapy and rehabilitation, tailored to the athlete.” Ref: https://doi.org/10.3390/medicina57111208
53 The paragraph is very informative and original, enriched with other bibliographical references.
If the approach is narrative I agree on a table, but from a methodological point of view it seems a purely systematic research, I would review it in two rows or columns recalling the previous concept of short-term or long-term. I would perhaps associate a figure
Table 2 as said previously
398 This review suggests .. it appears, it seems .. these definitive conclusions have been softened because there are few articles, among other things recently with the pandemic and with the resumption of the post-lockdown championships there has not been this wave of injuries as it was expected
Author Response
23 put football because it can enrich the dissemination of the manuscript
Thanks. Football has been sdded to the keyword.
30 I recommend more than 2 bibliographic references for such full-bodied statements. Is this number of injuries really greater?
Thanks for this great comment. We have added references there.
32 here too, have they really modified training management to address these concerns? If so, please provide bibliographic references.
Thanks. We have added references here.
37 This sentence can serve as a basis for the rationale for the study objective .. move it to the end of the introduction
Thank you very much for this suggestion. We have already moved it to the end of the introduction.
49 Maybe I can suggest a rationale like this: “In this scenario, the approach to these relationships has changed, with greater individualization of training, for pain management, physical therapy and rehabilitation, tailored to the athlete.” Ref: https://doi.org/10.3390/medicina57111208
Thank you very much for your constructive suggestion. We added this view and reference 67 in the middle of the 48-50.
53 The paragraph is very informative and original, enriched with other bibliographical references.If the approach is narrative I agree on a table, but from a methodological point of view it seems a purely systematic research, I would review it in two rows or columns recalling the previous concept of short-term or long-term. I would perhaps associate a figure.Table 2 as said previously
Thank you for your suggestion. Considering that some studies included both short-term congestion and long-term congestion simultaneously, as well as detailed comparisons in sentences, we didn’t put the short-term and long-term content in two tables.
398 This review suggests .. it appears, it seems .. these definitive conclusions have been softened because there are few articles, among other things recently with the pandemic and with the resumption of the post-lockdown championships there has not been this wave of injuries as it was expected
Thanks for pointing out this.We have revised the conclusion of the article to soften it.
Reviewer 2 Report
The present study is of interest to provide practical guidance for workload management and injury prevention in professional athletes.
This paper reviewed the literature on the effect of load on injury risk, injury prediction, and interpretation mechanisms in professional male soccer players.
Despite the interesting work and sound study, I would like to present some concerns/suggestions before proceeding with further revisions.
1. The topic and the paper it is very interesting, however, with all due respect, I am not so sure what this paper could add more to our scientific literature considering the information that is already available in many scientific databases (e.g: PubMed:https://pubmed.ncbi.nlm.nih.gov/34206948/; https://pubmed.ncbi.nlm.nih.gov/26822969/), including reviews and systematic reviews. So, in my opinion, this paper could be significantly improved if the authors could also add information regarding youth male and female and also professional female soccer players.
2. According to the above suggestions, I also would like to challenge the authors to increase the quality of the paper, by changing this review to a systematic review or even into a systematic review with meta-analyses. It is reasonable to think, that following the above suggestions it is possible to reach a more high audience when compared to the audience that will read the current paper, despite their interesting and sound study/topic.
Author Response
1. The topic and the paper it is very interesting, however, with all due respect, I am not so sure what this paper could add more to our scientific literature considering the information that is already available in many scientific databases (e.g: PubMed:https://pubmed.ncbi.nlm.nih.gov/34206948/;https://pubmed.ncbi.nlm.nih.gov/26822969/), including reviews and systematic reviews. So, in my opinion, this paper could be significantly improved if the authors could also add information regarding youth male and female and also professional female soccer players.
Thanks for this suggestion.We didn’t include literature on youth and female in our study for two reasons. First, there is very little research literature on professional female players. Second, the workload and the injury characteristics of youth players are significant different from adult players. We targeted adult male professional players in order to enhance the relevance and specificity of the study.
2.According to the above suggestions, I also would like to challenge the authors to increase the quality of the paper, by changing this review to a systematic review or even into a systematic review with meta-analyses. It is reasonable to think, that following the above suggestions it is possible to reach a more high audience when compared to the audience that will read the current paper, despite their interesting and sound study/topic
Thanks for pointing out this. We have modified this review to a systematic review to improve the quality of paper.
Round 2
Reviewer 1 Report
The work has been deepened and made more solid, thanks.
Author Response
Response to Reviewer: 1 The work has been deepened and made more solid, thanks. Thank you very very much for your encouragement. And we checked all the spells and references in our manuscript again.
Reviewer 2 Report
The authors did a good job on reviewing the manuscript and answering all the revisions maded.
Nevertheless, I ask the authors to update PRISMA to preferred reporting items for systematic reviews and meta-analyses (PRISMA) 2020 guidelines (https://www.bmj.com/content/372/bmj.n71).
Moreover, please present the preferred reporting items for systematic reviews and meta-analyses (PRISMA) diagram of the literature search results, and the item checklist as proposed by Page et al 2021.
Author Response
The authors did a good job on reviewing the manuscript and answering all the revisions maded. Thank you very very much for your encouragement. Nevertheless, I ask the authors to update PRISMA to preferred reporting items for systematic reviews and meta-analyses (PRISMA) 2020 guidelines (https://www.bmj.com/content/372/bmj.n71).
Thank you very much for your constructive suggestions. We have already updated the PRISMA 2020 in our manuscript.
Moreover, please present the preferred reporting items for systematic reviews and meta-analyses (PRISMA) diagram of the literature search results, and the item checklist as proposed by Page et al 2021.
Thank you very much for pointing this . We've already added flowchart of the search process and study selection (see Figure 1). Besides, we checked all the items and did some minor adjustments in the review according to the items of PRISMA2020.
